# Paramagnetic NMR Spectroscopy Is a Tool to Address Reactivity, Structure, and Protein–Protein Interactions of Metalloproteins: The Case of Iron–Sulfur Proteins

**Mario Piccioli** 

Magnetic Resonance Center and Department of Chemistry, University of Florence, 50019 Sesto Fiorentino, Italy; piccioli@cerm.unifi.it

**Abstract:** The study of cellular machineries responsible for the iron–sulfur (Fe–S) cluster biogenesis has led to the identification of a large number of proteins, whose importance for life is documented by an increasing number of diseases linked to them. The labile nature of Fe–S clusters and the transient protein–protein interactions, occurring during the various steps of the maturation process, make their structural characterization in solution particularly difficult. Paramagnetic nuclear magnetic resonance (NMR) has been used for decades to characterize chemical composition, magnetic coupling, and the electronic structure of Fe–S clusters in proteins; it represents, therefore, a powerful tool to study the protein–protein interaction networks of proteins involving into iron–sulfur cluster biogenesis. The optimization of the various NMR experiments with respect to the hyperfine interaction will be summarized here in the form of a protocol; recently developed experiments for measuring longitudinal and transverse nuclear relaxation rates in highly paramagnetic systems will be also reviewed. Finally, we will address the use of extrinsic paramagnetic centers covalently bound to diamagnetic proteins, which contributed over the last twenty years to promote the applications of paramagnetic NMR well beyond the structural biology of metalloproteins.

**Keywords:** iron–sulfur proteins; paramagnetic NMR; iron–sulfur cluster biogenesis; solution structures; metalloproteins

## 1. Introduction

Paramagnetic nuclear magnetic resonance (NMR) has been, over the last twenty years, one of more lively and active branches of biomolecular NMR, widely used to characterize metalloproteins. Indeed, metalloproteins represent a wide percentage of the entire proteome and a large share of metalloproteins is paramagnetic. The first solution structure of a paramagnetic metalloprotein was solved in 1994 [1]. Since then, many protein structures of paramagnetic systems were obtained in solution in different oxidation states [2–6]. The quest for novel methodological advancements flourished, redox dependent effects were investigated [7], and a number of paramagnetism-based NMR restraints were proposed [8]. Likewise, the main programs for solution structures calculations were revisited and complemented with routines able to tackle paramagnetism-derived NMR restraints and to combine them with conventional NMR restraints [9,10]. Over the last decade, the popularity of NMR-based structural biology has been shadowed by a number of factors: the "mild" success of NMR as a high-throughput method for protein structure determination, the increasing performances of structure prediction approaches, the appearance into the scene of the brilliant and raising star of cryo-electron microscopy, that is replacing NMR for an increasing number of applications in proteomics and interactomics and is integrated with NMR to obtain structure determination of very large complexes at an atomic resolution [11,12]. Nevertheless, many methodological developments have been proposed

that have contributed to expand the range of applications [13–18]; within this scenario, the exploitation of the hyperfine interaction has been one of the most exciting aspects.

In paramagnetic metalloproteins, the hyperfine interaction between electron spin and nuclear spins can be a tool to: (i) elucidate catalytic mechanisms in metalloenzymes and provide a molecular picture of the currently known protein–protein interaction networks involving metalloproteins; (ii) use small and stable metalloproteins as test systems to develop novel experiments and to obtain additional NMR restraints that could eventually be used to study larger and unstable proteins. However, probably the most intriguing aspect is the use of metal-based spin labels as an additional source of structural constraints in diamagnetic proteins. This succeeded to extend the range of systems that can be studied via paramagnetic NMR: extrinsic paramagnetic centers contributed to promote the applications of paramagnetic NMR also beyond structural biology in solution [19–25].

One of the aspects to which paramagnetic NMR has substantially contributed is the discovery of molecular machineries devoted to the biogenesis of iron sulfur proteins and the study of cellular trafficking of metal cofactors. I will briefly overview the contribution of NMR studies for elucidating aspects of iron–sulfur proteins biogenesis, where the understanding at a molecular level provide snapshots of protein–protein interactions, which are crucial for the biomedical aspects. Then, I will present here a summary of the recent developments in NMR methodologies for paramagnetic proteins and show how they can be used within solution structure calculations. Finally, I will briefly overview how paramagnetic NMR came under the spotlights when extrinsic paramagnetic agents have been attached to biomolecules and used as a source of paramagnetism based NMR restraints.

## 2. Paramagnetic NMR

The theory of the hyperfine interaction between electron spins and nuclear spins and its consequences on the nuclear relaxation properties have been exhaustively reviewed [2,26]. For the ease of the reader, I will recap here the terms that are of major use in paramagnetic systems. The hyperfine shift, i.e., the contribution to the chemical shift arising from the hyperfine interaction, can be factorized into a contact (*CS*) and pseudo-contact (*PCS*) contributions, according to Equations (1)–(3)

$$\partial_{obs} = \partial_{CS} + \partial_{PCS} \tag{1}$$

$$\partial_{CS} = \frac{A}{\hbar} \frac{\overline{g} \mu_B S(S+1)}{3 \gamma_I kT} \tag{2}$$

$$\partial_{PCS} = \frac{1}{12\pi r_{MI}^3} \left[ \Delta\chi_{ax}^{para}(3\cos^2\theta_{MI} - 1) + \frac{3}{2}\Delta\chi_{rh}^{para}(\sin^2\theta_{MI}\cos 2\varphi_{MI}) \right] \tag{3}$$

where $A$ is the hyperfine coupling constant, which is proportional to the electron spin density at the nucleus and can be anisotropic due to electron orbital contributions; $g$ is the average $g$ value along the principal directions of the contact coupling, when the latter is anisotropic; $\mu_B$ is the electron Bohr magneton; $S$ is the electron spin number; $\gamma_I$ is the gyromagnetic ratio of a generic I nucleus; $k$ is the Boltzmann constant; $T$ is the absolute temperature; $r_{MI}$ is the distance of the nucleus $I$ from the metal ion $M$; $\Delta\chi_{ax}^{para}$ and $\Delta\chi_{rh}^{para}$ are the axial and rhombic components of the anisotropic magnetic susceptibility tensor; $\theta_{MI}$ and $\varphi_{MI}$ are the polar angles of the nucleus I with respect to the principal axes of the magnetic susceptibility anisotropy tensor in a reference system that has M in its origin.

As it comes from Equations (1)–(3), contact shift (*CS*) is operative wherever the nucleus experiences unpaired electron spin density, which occurs through direct spin delocalization and/or spin polarization. The contact contribution is different from zero only for the nuclei of the ligands of the paramagnetic metal ion(s) or groups interacting with them by H-bonds. *CS* can be useful to obtain information on the dihedral angles of residues coordinating the metal center [27,28]. The *PCS* term is operative when the paramagnetic center gives rise to a magnetic anisotropy tensor, and due to the $r^{-3}$ dependence, it may be effective on nuclei that are up to 40 Å apart from the paramagnetic center [29]; it is structure-dependent and can eventually be converted into a structure restraint [9].

The contributions to paramagnetic relaxation are summarized in Equations (4)–(9)

$$R_{1,2} = R_{1,2}^{dia} + R_{1,2}^{para} \tag{4}$$

$$R_{1Cont} = \frac{2}{3}\left(\frac{A}{\hbar}\right)^2 S(S+1)\frac{\tau_c}{1+(\omega_I-\omega_S)^2\tau_c^2} \tag{5}$$

$$R_{1Dip} = \frac{2}{15}\left(\frac{\mu_0}{4\pi}\right)^2 \frac{\gamma_I^2\mu_B^2 g_e^2 S(S+1)}{r_{MI}^6}\left(\frac{\tau_c}{1+(\omega_I-\omega_S)^2\tau_c^2} + \frac{3\tau_c}{1+\omega_I^2\tau_c^2} + \frac{6\tau_c}{1+(\omega_I+\omega_S)^2\tau_c^2}\right) \tag{6}$$

$$R_{2Cont} = \frac{1}{3}\left(\frac{A}{\hbar}\right)^2 S(S+1)\left(\frac{\tau_c}{1+(\omega_I-\omega_S)^2\tau_c^2} + \tau_c\right) \tag{7}$$

$$R_{2Dip} = \frac{1}{15}\left(\frac{\mu_0}{4\pi}\right)^2 \frac{\gamma_I^2\mu_B^2 g_e^2 S(S+1)}{r_{MI}^6}\left(4\tau_C + \frac{\tau_c}{1+(\omega_I-\omega_S)^2\tau_c^2} + \frac{3\tau_c}{1+\omega_I^2\tau_c^2} + \frac{6\tau_c}{1+(\omega_I+\omega_S)^2\tau_c^2} + \frac{6\tau_c}{1+\omega_S^2\tau_c^2}\right) \tag{8}$$

$$R_{2Curie} = \frac{1}{5}\left(\frac{\mu_0}{4\pi}\right)^2 \frac{\omega_I^2\mu_B^4 g_e^4 S^2(S+1)^2}{(3kT)^2 r_{MI}^6}\left(4\tau_C + \frac{3\tau_c}{1+\omega_I^2\tau_c^2}\right) \tag{9}$$

where $\omega_I$ and $\omega_S$ are the Larmor frequencies of the nuclear and electron spins $I$ and $S$, respectively, $g_e$ is the free electron g value and the other symbols are defined above. The correlation time for the interactions contributing to the relaxation is indicated by $\tau_c$ and represents the result of various dynamic factors modulating the nuclear-electron spin interaction, each of them characterized by its own correlation time. The relaxation process with the shortest $\tau$ becomes dominant, as indicated in Equation (10):

$$\tau_c^{-1} = \tau_s^{-1} + \tau_r^{-1} + \tau_M^{-1} \tag{10}$$

where $\tau_s$ is the electron relaxation correlation time, $\tau_r$ is the rotational correlation time and $\tau_M$ is the exchange correlation time that is operative only in the presence of chemical or conformational equilibria. Although in Equations (3)–(9) the same symbol ($\tau_c$) has been used, in the case of contact relaxation (Equations (5) and (7)), only chemical exchange ($\tau_M$) and electron relaxation ($\tau_s$) can modulate the coupling between the electron and the nucleus, while in Curie spin relaxation (Equation (9)) only chemical exchange ($\tau_M$) and rotational correlation ($\tau_r$) are effective. Dipolar and Curie relaxation mechanisms have a r$^{-6}$ dependence from the metal-to-nucleus distance. This allows the use of paramagnetic relaxation rate enhancement (PRE) as a source of long-range distance restraint, as it will be illustrated in Section 5.

With the advent of high-field NMR spectrometers, other paramagnetic effects have become observable and measurable. Among them, the most popular effect, which however does not arise from the hyperfine interaction, is the partial orientation along the magnetic field, giving rise to non-completely averaged, i.e., residual, dipolar couplings (*RDC*s) [30]. In diamagnetic systems, this is accomplished by dissolving the molecules in anisotropic solvent or orienting media [31]. In paramagnetic proteins, when the metal center has non-zero magnetic susceptibility anisotropy, the molecular magnetic susceptibility is dominated by the paramagnetic contribution ($\chi_{para}$), which is the same magnetic susceptibility responsible for *PCS*. As reported in Equation (11) for the case of the $^{15}$N of backbone amide and its attached proton, the paramagnetic residual dipolar coupling (*RDC*$^{PARA}$) for a pair of nuclei depends on the orientation of the vector connecting the two nuclei with respect to the magnetic susceptibility tensor axes (but not, as with *PCS*, on the distance from the paramagnetic center):

$$RDC^{PARA} = \frac{1}{4\pi}\frac{B_0^2}{15kT}\frac{\gamma_H\gamma_N h}{4\pi^2 r_{NH}^3}\left[\Delta\chi_{ax}^{para}(3\cos^2\theta_{NH}-1) + \frac{3}{2}\Delta\chi_{rh}^{para}(\sin^2\theta_{NH}\cos 2\varphi_{NH})\right] \tag{11}$$

where $r_{NH}$ is the distance between the amide proton and the amide nitrogen and is generally considered fixed; $\theta_{NH}$ and $\varphi_{NH}$ are the polar angles that describe the orientation of the inter-nuclear N–H vector with respect to the alignment tensor and all the other symbols have the usual meanings.

As far as relaxation rates are concerned, Equations (5)–(9) do not take into account the occurrence of cross correlation effects between different relaxation mechanisms modulated by the same correlation time that produce many potentially relevant effects [32,33]. In paramagnetic system, the most widely exploited cross-correlated effect are the cross correlation rates (*CCR*) between Curie relaxation and dipolar coupling. Considering again the coupling between $^{15}$N of backbone amide and its attached proton, the equation is

$$CCR = \frac{2}{15\pi}\left(\frac{\mu_0}{4\pi}\right)^2 \frac{B_0 \gamma_H^2 \gamma_N \mu_B^2 g_e^2 \hbar S(S+1)}{r_{NH}^3 r_{MH}^3 kT}\left(4\tau_C + \frac{3\tau_c}{1+\omega_0^2 \tau_c^2}\right)\frac{(3\cos^2\varphi_{CCR}-1)}{2} \tag{12}$$

where the only new symbol $\varphi_{CCR}$ is the angle between the metal–proton vector and the N–H vector. *CCR* are relevant when the Curie spin relaxation from Equation (9) becomes the dominant contribution to transverse relaxation, as it typically occurs for systems with large S values and high molecular mass and at very high magnetic fields. Finally, for paramagnetic systems characterized by strong magnetic anisotropy, the scenario can be further complicated: a substantial angular dependence of nuclear relaxation rates has been observed due to relaxation anisotropy [34,35], while significant additional contribution to relaxation have been observed from other cross correlation mechanisms, such as those involving Curie spin relaxation and chemical shift anisotropy [36]. Moreover, in several cases a substantial deviation from the r$^{-6}$ dependency of PREs has been observed and interpreted as due to non-specific intermolecular PREs [37].

## 3. Iron–Sulfur Proteins: From Electron Transfer to Cluster Biogenesis

Iron–sulfur proteins have been one the first class of metalloproteins actively studied using NMR spectroscopy tailored to paramagnetic systems (nowadays commonly termed as "paramagnetic NMR"). Paramagnetic NMR contributed to the analysis of the magnetic coupling patterns [38–41] and to the understanding of the electronic structure of [2Fe–2S], [3Fe–4S], and [4Fe–4S] clusters bound to small electron transfer proteins. In many cases, both oxidation states involved in electron transfer processes were investigated and the structural and spectroscopic differences between them related to protein structure–function relationships [42,43]. Electron transfer proteins that have been studied during the 1990s were small, soluble, and thermodynamically stable, such as rubredoxins, ferredoxins, and high potential iron–sulfur proteins [5,44–56]. These proteins acted as model systems for the study of more complex cases, recently isolated and characterized, in which transient sites, conformational flexibility, and protein–protein interactions made the NMR investigation more challenging [57–59]. Indeed, the NMR characterization of metalloproteins involved in metal homeostasis and trafficking stimulated new methodological developments but also the revival of old NMR approaches [43,60].

In the last decade, the iron–sulfur cluster assembly (ISC) in mitochondria and the cytosolic iron–sulfur assembly system (CIA) machineries have been extensively studied via in vivo assays and genetic approaches [61,62]. The combination of solution NMR standard experiments with those tailored to paramagnetic systems has often been crucial to characterize, at the molecular level, the interaction networks responsible for maturing human mitochondrial and cytosolic Fe–S proteins. The human proteins of the iron–sulfur cluster (ISC) assembly machinery are all soluble proteins located in the mitochondrial matrix [63]. *De novo* Fe–S cluster synthesis occurs on the mitochondrial scaffold protein ISCU and requires a high molecular weight complex composed of five proteins: frataxin, a protein for which several functions have been proposed [64–70], the enzyme cysteine desulfurase NFS1, the small factors ISD11 and acyl carrier protein (ACP), and ISCU [71,72]. In vivo data [73] showed that the second step of the human ISC assembly process is the transfer of the newly synthetized cluster to the mitochondrial monothiol glutaredoxin, GLRX5, which acts as a [2Fe–2S] cluster transfer protein.

Then, the [4Fe–4S] assembly process occurs, involving the interaction of [2Fe–2S]-GLRX5 with two homologous proteins, ISCA1 and ISCA2, with another protein, IBA57. Mitochondrial [4Fe–4S] protein assembly involves reductive [2Fe–2S] cluster fusion on ISCA1–ISCA2 by electron flow from ferredoxin FDX2 [74].

NMR has largely contributed to describe how two [2Fe–2S] clusters couple each other to form a [4Fe–4S] cluster. Solution structures of both apo- and holo-GLRX5 clearly show that apo-GLRX5 is monomeric in solution and it undergoes dimerization only upon cluster binding [75]. This is at variance with the crystal structure that reports, for [2Fe–2S]-GLRX5, a homo-tetramer where two [2Fe–2S] clusters are coordinated by four protein subunits and four GSH molecules [76]. By mapping the chemical shift variations between apo- and holo-GLRX5 it has been found that the protein region affected by cluster binding involves a 10 Å radius sphere centered on the [2Fe–2S] cluster, while the latter is bridging the two subunits of the symmetric dimer [75]. Detailed enough to describe the overall conformation in solution, routine NMR experiments are not sufficient to describe the proximity of the cluster: the backbone NH signals of 11 residues located inside this sphere were not detected in the standard $^1$H–$^{15}$N heteronuclear single quantum coherence (HSQC) experiment. Methodological developments, illustrated in the next section, allowed us to revive these signals and to identify, for the residues close to the [2Fe–2S] cluster, two sets of signals, thus demonstrating that dimer [2Fe–2S]-GLRX5 exists in solution as a mixture of two species in equilibrium. The structural plasticity of the dimer state of [2Fe–2S]-GLRX5 is the crucial factor that allows an efficient cluster transfer to the partner proteins human ISCA1 and ISCA2 via a specific protein–protein recognition mechanism.

The interaction of holo-GLRX5 with the proteins ISCA2 and ISCA1, the transfer of the [2Fe–2S] cluster from GLRX5 to the receiving proteins and the formation of the [4Fe–4S] can be successfully characterized by mapping, during protein–protein interaction experiments, chemical shift variations of backbone HN groups via $^{15}$N HSQC. It was shown that cluster transfer occurs uni-directionally from GLRX5 to apo-ISCA1 and ISCA2, and that only one of the two conformations of [2Fe–2S]-GLRX5 previously identified is responsible of the cluster transfer and that both ISCA1 and ISCA2 receive the [2Fe–2S] cluster in their dimeric states. In this case, "old fashioned" one-dimensional paramagnetic NMR spectra provided a clear picture of the interaction and, supported by electrospray ionization—mass spectrometry (ESI-MS) and electron paramagnetic resonance (EPR) spectra, were essential to monitor the formation of the $[Fe_4S_4]^{2+}$ cluster [77]. Indeed, the NMR spectra of Figure 1 showed that, while the "as purified" holo-ISCA2 protein has a NMR spectrum typical of a $[Fe_2S_2]^{2+}$ cluster-containing species, the chemical reconstituted holo-ISCA2 protein is predominantly a [4Fe–4S] bound, dimeric species with only a minor component of $[Fe_2S_2]^{2+}$ bound-cluster.

The number of signals and their temperature dependences indicate that the [4Fe–4S] is in the reduced $[Fe_4S_4]^+$ state and that holo-ISCA2 exists in solution in at least two different conformations, characterized by different coordination environments around the $[Fe_4S_4]^+$ cluster. Paramagnetic NMR gives us insights on how the structural properties of the protein drive the electronic structure of the inorganic cofactor. The electronic distribution within the $[Fe_4S_4]^+$ cluster was found to be different from the one observed in bacterial ferredoxins [44]. In holo-ISCA2, the $[Fe_4S_4]^+$ cluster is at the interface of two identical monomers, the scaffold around the cluster is highly symmetric thus making the four iron ions of the cluster essentially equivalent. The equivalence among the iron ions produces a large electron delocalization, which decreases the effective J values among the iron ions, determining downfield shifts and anti-Curie temperature dependence for signals from all Cys βCH$_2$, that are shown in Figure 1A, upper part. In the case of ferredoxins, the inequivalence of the four iron sites and the magnetic coupling gave a purely ferrous iron ion pair ($Fe^{2+}$-$Fe^{2+}$) and a mixed valence ($Fe^{2.5+}$-$Fe^{2.5+}$) pair. This provided the possibility to assign Cys βCH$_2$ signals as bound to the purely ferrous or to the mixed valence iron ion pair according to their temperature dependence [44].

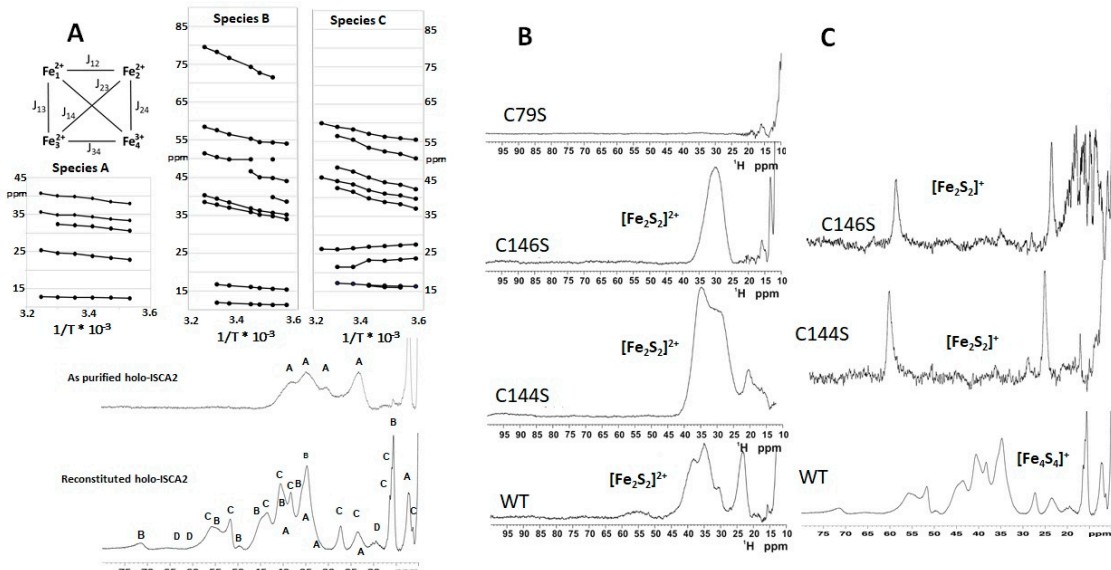

**Figure 1.** (**A**) Lower part: $^1$H NMR spectra of "as purified" holo-ISCA2 and of chemically reconstituted holo-ISCA2 in 50 mM phosphate buffer pH 7.0 at 600 MHz and 283 K. Signals labeled A arise from Hβ and Hα from cysteine ligands coordinated to a $[Fe_2S_2]^{2+}$ cluster. Signals labeled B, C, and D arise from $H_\beta$ and $H_\alpha$ from cysteine ligands coordinated to a $[Fe_4S_4]^+$ cluster in different species that have been identified according to relative intensities of signals B−D observed in different reconstituted protein samples. Upper part: Temperature dependence of the chemical shifts for the hyperfine-shifted signals of the three different species (A, B, C) of chemically reconstituted holo-ISCA2. Experiments were recorded at 600 MHz, pH 7.0 in the temperature range of 280–308 K. A schematic representation of a [4Fe–4S] cluster and its coupling scheme in the reduced state $[Fe_4S_4]^+$ is shown. (**B**) $^1$H NMR spectra of chemically reconstituted ISCA2 mutants C79S, C144S, C146S and of wild-type $[Fe_2S_2]^{2+}$–ISCA2 purified from E. coli in 50 mM phosphate buffer pH 7.0 at 600 MHz and 283 K. (**C**) $^1$H NMR spectra of the same samples as in panel B, recorded upon addition of a 5mM dithionite as reducing agent. Figure adapted from References [77,78].

The mechanism of the formation of the [4Fe–4S] cluster in the ISCA1–ISCA2 hetero-dimer complex and in the ISCA2–ISCA2 homo-dimer complex can be understood when Cys-to-Ser single mutants for each conserved cysteine of ISCA2 were studied, in order to monitor the cluster transfer from $[Fe_2S_2]^{2+}$-GLRX5. ISCA2 has three cysteine residues and, because the holo-ISCA2 is a dimer with a [4Fe–4S] cluster at the interface, we expect that two out of three cysteine residues bind the cluster in a symmetric dimeric fashion. In ISCA2, two of the three Cys belongs to the C-terminal site (Cys 144 and Cys 146) while the third (Cys 79) is within a structured protein region. In addition, in this case, 1D paramagnetic NMR was crucial to demonstrate the different roles of these cysteine residues in the cluster transfer process [78]. As shown in Figure 1B, the C79S mutant does not bind the cluster, clearly indicating that C79 has a crucial role in stabilizing the holo- form of ISCA2. However, Cys 79 is not involved in the cluster transfer step, as the C79S ISCA2 mutant is still able to extract the [2Fe–2S] cluster from GLRX5. The mutations of either C144S or C146S, i.e., the two C-term cysteines, give rise to the formation of $[Fe_2S_2]^{2+}$–ISCA2 adducts. Upon reduction, both mutants give rise to $[Fe_2S_2]^+$–ISCA2 species without any evidence of the formation of a [4Fe–4S] cluster, thus indicating that the [4Fe–4S]–ISCA2 derivative is formed only when all the three Cysteine residues are present. According to the scheme summarized in Figure 2, it was proposed that the species coordinating the cluster with the C-terminal cysteines can evolve into a more thermodynamically favored species, which binds the [2Fe–2S] cluster in the dimer by Cys 79 and either Cys 144 or Cys 146.

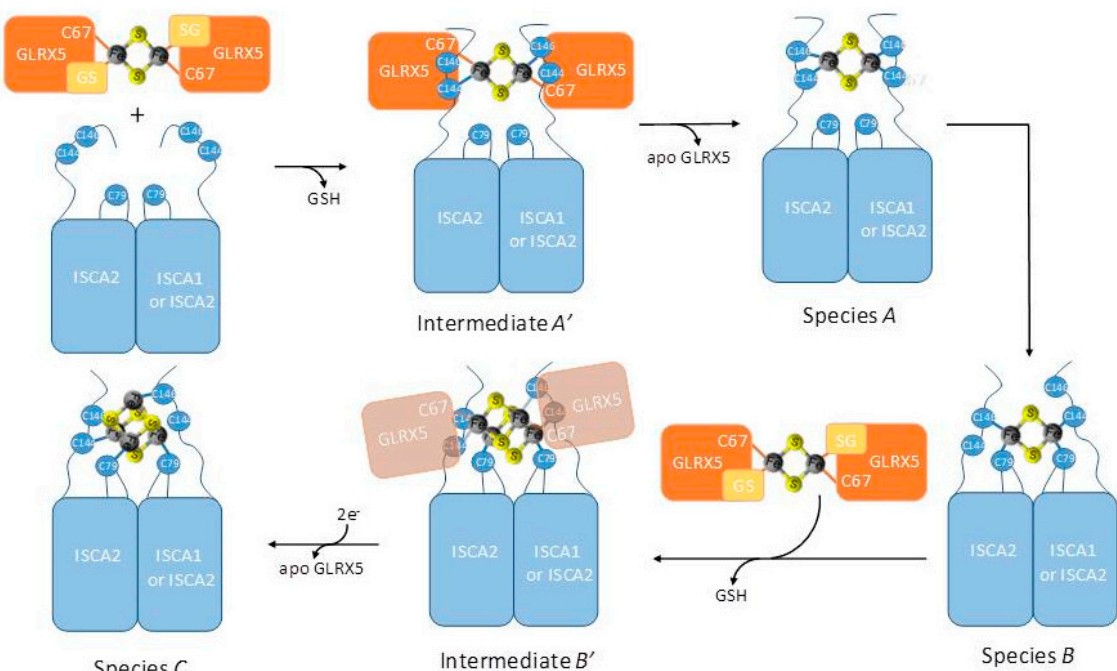

**Figure 2.** Model of the [4Fe–4S] cluster assembly mechanism by holo-GLRX5, apo-ISCA1/ISCA2 interactions. Adapted from Reference [78].

This mechanism would also make two of the C-terminal cysteine residues (Cys 144 and Cys 146 of ISCA2) available for the coordination of a second cluster, which can be extracted from GLRX5 with the formation of another GLRX5–ISCA2 intermediate. This transient intermediate, which contains two $[Fe_2S_2]^{2+}$ clusters, might be the species that, by accepting two electrons from a physiological electron donor recently identified as FDX2 [74], evolves to the final [4Fe–4S]–ISCA2 complex. A reductive coupling of two $[Fe_2S_2]^{2+}$ clusters, which is a general mechanism for generating a [4Fe–4S] cluster [79,80], would therefore occur on the latter transient intermediate to form a $[Fe_4S_4]^{+}$ cluster bound to the dimer.

## 4. New Experiments in NMR of Paramagnetic Molecules and Their Applications to Iron–Sulfur Proteins

In iron–sulfur proteins, significant hyperfine shifts are observed only for a few signals of cluster-bound cysteine residues, because the pseudo-contact term of the hyperfine shift is almost negligible and only the contact contributions, arising from the spin delocalization from the cluster to the iron-bound residues, are observed [81]. However, paramagnetic relaxation is dominated by dipolar contributions; it is essentially driven by the electron relaxation times of the iron ions of the cluster and shows a $r^{-6}$ dependency from the metal-to-atom distance [26]. The resulting picture is that many signals, which do not belong to the cluster-bound residues but are close to the paramagnetic center, are broadened by paramagnetic relaxation but not shifted outside the bulk diamagnetic envelope. Large contributions to relaxation and small contributions to chemical shift represent the most challenging situation for resonance assignment [82]; this is the reason why iron–sulfur proteins are challenging and often used as paradigmatic cases for the development of novel NMR experiments and for the optimization of relaxation based NMR restraints.

The optimization of the various NMR experiments can be summarized in the form of a protocol describing the steps that are required when NMR experiments are tailored for paramagnetic systems: (i) remove from a sequence all "un-necessary" steps that involve $^1$H transverse relaxation; (ii) adjust all the experimental parameters such as number of scans, spectral windows, acquisition, recycle, and coherence transfer delays according to the relaxation properties of the concerned coherences; (iii) for polarization transfer based experiments, change the detection scheme by replacing the in-phase

acquisition with the antiphase acquisition and maximize the efficiency of the coherence transfer, which critically depends on nuclear relaxation; (iv) add relaxation based filters that can be tuned according to the range of $T_1$ and $T_2$ values of interest; (v) exploit the detection of nuclei with low gyromagnetic ratio, using dedicated hardware.

Under favorable conditions, this approach succeeded to remove the blind sphere around a paramagnetic center and provided a complete resonance assignment for paramagnetic metalloproteins [83,84]. Based on the above criteria, triple-resonance experiments such as HNCA, HNCO, and CBCANH, characterized by many pulses and many polarization transfer steps, can be efficiently optimized for the identification of paramagnetic signals. Figure 3A shows the effect of implementations on a CBCA(CO)NH experiment, considering a routine version of the pulse sequence [85]. The coherence transfer efficiency for a fast relaxing signal is affected by the removal, within the sequence, of some of the building blocks that are routinely used in triple-resonance experiments such as crush gradients, echo-antiecho detection, and sensitivity improvement. The removal of these building blocks provides a substantial improvement of the coherence transfer efficiency for paramagnetic fast relaxing signals. This optimization strategy can be easily applied to all available triple-resonance experiments, allowing one to record "paramagnetic" versions of, virtually, any triple-resonance experiment. As shown in Figure 3B, an HNCA optimized as above permits the identification of the HNCA peaks of three out of the four cluster-bound cysteine residues that, by no means, are observable using a routine HNCA setup [86].

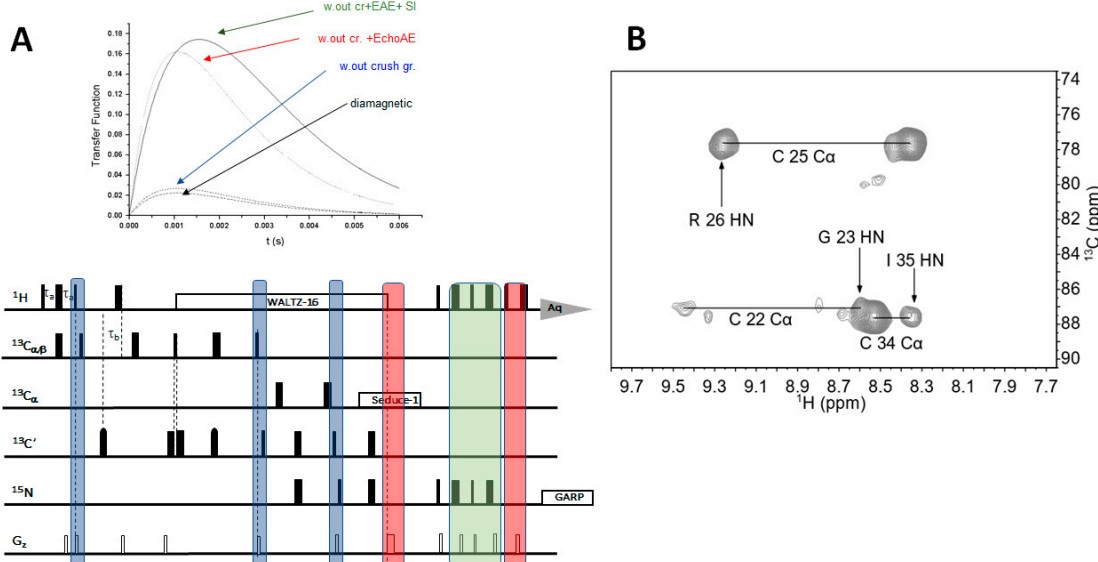

**Figure 3.** (**A**) Lower part: pulse scheme for a routine CBCA(CO)NH experiment [87]. Blocks highlighted in blue are the crush gradients that can be removed without affecting the coherence transfer pathways of the sequence, those in red correspond to the echo-antiecho blocks, and the one in green refers to the sensitivity improvement part of the reverse INEPT. Upper part: Calculated transfer functions for the NH reverse INEPT transfer for a $^1$H signal with $R_2 = 600$ s$^{-1}$, $R_1 = 120$ s$^{-1}$, considering $\tau_e = 5 \times 10^{-10}$ s and $\tau_r = 6 \times 10^{-9}$ s [85]. All transfer functions are normalized with respect to a normal reverse INEPT under optimized condition for the transfer delay and neglecting losses due to $^1$H-$^{15}$N relaxation. (**B**) 500 MHz 298 K, H(N)CA spectrum of HiPIP protein PioC optimized for peaks involving fast relaxing resonances. The experiment allows identification and assignment of three out of the four cluster-bound cysteines in the HNCA. Adapted from Reference [86].

For the $^{15}$N-HSQC, the typical fingerprint experiment in protein NMR, the optimization of INEPT, and reverse INEPT delays may not be sufficient to identify residues at short distance from a paramagnetic center. The removal of the inverse INEPT coherence transfer step and the insertion of

an inversion recovery (IR) filter prior to the $^1$H 90° start pulse contributed to the detection of highly paramagnetic signals. The sequence, shown in Figure 4A, is called IR-HSQC-AP [88]. The reverse INEPT block is removed to avoid signal losses due to $^1$H $R_2$ relaxation. The $2H_yN_z$ coherence, created by the two 90° pulses at the end of $^{15}$N evolution, is acquired in antiphase (AP) without $^{15}$N decoupling. A dispersion phase mode of the antiphase doublets produces the sum of the two dispersive components of opposite phase, thus giving rise to a pseudo-singlet with the maximum of signal intensity [89]. The IR filter preceding the first INEPT edits $^1$H signals according to their relaxation rates: a suitable choice of the inter-pulse delays causes a sign discrimination between fast relaxing signals and slow relaxing signals [88].

The $^1$H–$^{15}$N IR-HSQC-AP was the key tool to identify protein–protein interacting regions close to the paramagnetic Fe–S cluster in two very interesting cases. In the [2Fe–2S] GLRX5, already mentioned in the previous section, the spectrum shown in Figure 4A succeeded to observe nine of the eleven residues located inside a 10 Å sphere from the cluster, which were not detected in the standard $^1$H–$^{15}$N HSQC experiment. Two sets of signals, observed for the Fe–S ligand Cys 38 and for Ser 41, with chemical shifts different from those of the apo-protein, indicated that dimer [2Fe–2S] GLRX5 exists in solution as a mixture of two species. Another application of the IR-HSQC-AP was the NMR characterization of the protein anamorsin. Anamorsin belongs to the human Cyotosolic iron–sulfur cluster assembly machinery (CIA), it is a multi-domain protein (312 amino acids) characterized by a well folded N-term domain, an unstructured linker of about 50 amino acid residues, and a C-term region, called CIAPIN1 domain, that binds two [2Fe–2S] clusters [90,91] and receives electrons from a diflavin reductase Ndor1 [92]. The 108 amino acid CIAPIN1 domain is largely unstructured and about 30% of the HN resonances are unobserved in a standard HSQC experiment due to paramagnetic line broadening. Thanks to the IR-HSQC-AP, thirteen resonances previously unobserved could be identified and assigned, and the measured $^1$H $R_1$ values were used as restraints to define the environment of the cluster in the CIAPIN1 domain [93]. When the inter-pulse delay of the inversion recovery block is arrayed, the IR-HSQC-AP experiment can be also used to obtain $R_1$ rates of $H_N$ signals, including those signals strongly affected by the hyperfine interaction. This is one of the most interesting features of the sequence: relaxation-based restraints can be obtained via $^1$H $T_1$ and $T_2$ measurements from $^{15}$N HSQC-type experiments. In this scenario, the IR-HSQC-AP is a very useful approach to increase the number of available $T_1$ values that can be converted into PREs.

Very recently, another modified HSQC scheme appeared that provides transverse relaxation rates $R_2$ in highly paramagnetic systems. To date, the most widely used experiment [94,95] provides very accurate $R_2$ rates for $R_2 \leq 50$ s$^{-1}$, however this approach loose accuracy in the range 50–90 s$^{-1}$ and it is not applicable for $R_2 > 100$ s$^{-1}$. In the $^1$H $R_2$-weighted $^{15}$N-HSQC-AP experiment [96], shown in Figure 4B, the relaxation delay is embedded within the INEPT evolution, the refocusing INEPT is removed and signal is acquired as an antiphase doublet as soon as the $H_yN_z$ magnetization is created by the last $^1$H 90° pulse. In this very simple sequence, $^1$H transverse relaxation is active only during the delay $T$, when the $2H_xN_z$ coherence evolves from $H_y$ as $H_xN_z\sin(\pi J_{HN}T)$. $R_2$ values up to 400 s$^{-1}$ could be measured, with a significant improvement with respect to a standard experiment, as shown in the reported spectrum. The implementation of these approaches to $^{13}$C HSQC experiments is straightforward and gives the possibility to measure $R_1$ and $R_2$ rates of non-exchangeable protons that are affected by paramagnetic relaxation but are not shifted out from the bulk diamagnetic envelope. We can therefore obtain an extended series of $R_1$ and $R_2$ values that can eventually be used as structural restraints for all those cases in which nuclear Overhauser effects (NOE) are missing or ambiguous.

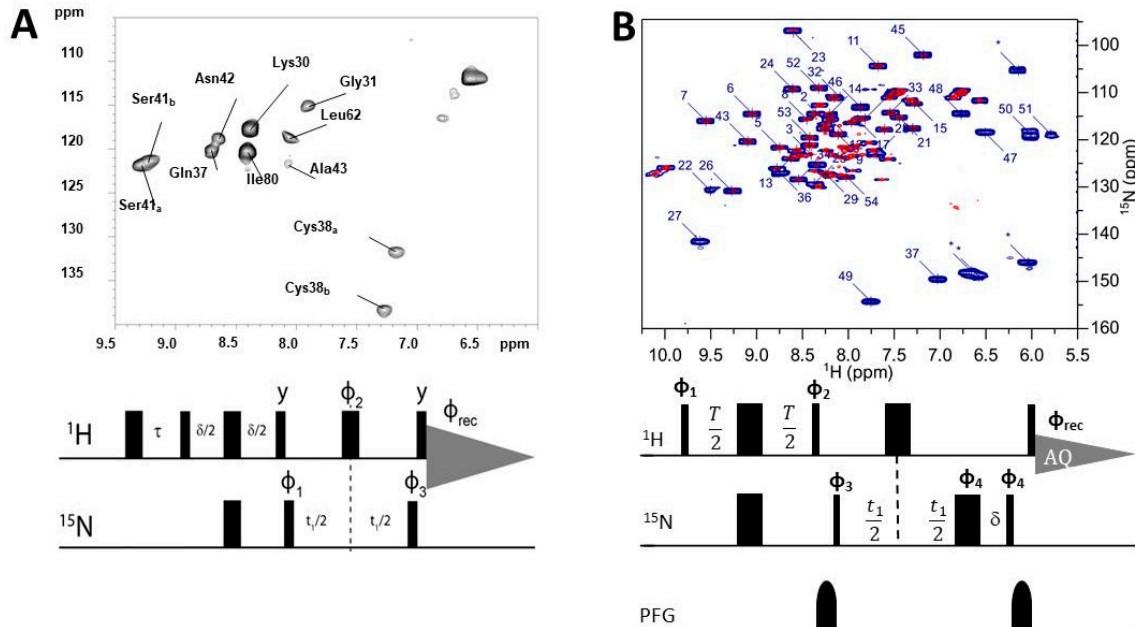

**Figure 4.** (**A**) Lower part: Pulse scheme of IR-$^{15}$N-HSQC-AP experiment. Phases as follows: $\varphi_1 = (x,-x)$, $\varphi_2 = (x,x,-x,-x)$, $\varphi_3 = (x,x,x,x,-x,-x,-x,-x)$, $\varphi_{rec} = (x,-x,x,-x,-x,x,-x,x)$. When undefined, phase x is used. Upper part: the IR-HSQC-AP spectrum of 1 mM holo-GLRX5 at 800 MHz is displayed reporting the paramagnetic NH signals and their assignment. The residues having two sets of signals in holo-GLRX5 are indicated with labels a and b. (**B**). Lower part: Pulse scheme of $^1$H $R_2$-weighted $^{15}$N HSQC-AP experiment. Phases as follows: $\varphi_1 = x,-x,y,-y$; $\varphi_2 = 2(y),2(x)$; $\varphi_3 = 2(x), 2(-x)$; $\varphi_4 = 4(x),4(-x)$; $\varphi_{rec} = x,-x,-x,x,-x,x,x,-x$. When undefined, phase x is used. Upper part: $^{15}$N HSQC spectra of PioC obtained with an $R_2$-weighted $^{15}$N-HSQC-AP experiment (blue) vs. the first point of the $R_2$ series collected with the standard experiment (red). Experiments were recorded using a 500 MHz. Folded peaks are marked with an asterisk. Figure adapted from References [75,88,96].

　　Advancements in NMR spectroscopy are invariably associated with developments of the available NMR instrumentation. A proof of this concept is the development of $^{13}$C direct detection methods, which constitute a major, well documented, achievement [97–100]. Cryogenically cooled probes optimized for $^{13}$C direct detection provided a significant improvement in the S/N achievable in $^{13}$C detected experiments. This has stimulated several groups to develop NMR approaches based on the direct detection of nuclei with low gyromagnetic ratios: $^{13}$C and $^{15}$N. Under many circumstances, such as unfolded proteins, proline-rich systems, or chemical exchange, $^1$H$_N$ signals may not be visible and the direct detection of $^{13}$C and/or $^{15}$N allows the circumventing the problem. This method is also particularly helpful for paramagnetic systems because, unlike the hyperfine shift, paramagnetic relaxation is dependent on $\gamma^2$: when passing from $^1$H- to $^{13}$C-detected experiments, the paramagnetic contributions to relaxation will be scaled by a factor $(\gamma_H/\gamma_C)^2$. This has been first applied in copper proteins [101,102] and in Ln(III)-substituted Calcium binding proteins [29,103] where $^{13}$C detection significantly reduced the blind sphere due to paramagnetism around the metal ion. It has also been shown that $^{13}$C protonless experiments significantly improve the detectability of effects such as residual dipolar couplings (*RDC* hereafter) involving fast relaxing $^1$H spins [104] as well as multiple quantum relaxation rates [105]. For iron–sulfur proteins, Markley and coworkers pioneered the idea and extensively used not only $^{13}$C, but also $^{15}$N and $^2$H direct detection, to obtain hyperfine shifts of heteronuclei when the corresponding $^1$H signals were broadened beyond detection for rubredoxins and also for Rieske proteins [52,53,100,106]. Among the very many $^{13}$C-detected experiments developed, the $^{13}$C–$^{13}$C correlation spectroscopy (COSY) [107,108] and the CACO (and COCACO) [109] were specifically tailored to paramagnetic systems. In both cases, the choice of $t_{1max}$ and $t_{2max}$ values can be

tuned according to the relaxation properties of the system; a comparison among two spectra recorded with different experimental parameters may be sufficient to identify residues in proximity of the paramagnetic center [110].

## 5. Paramagnetism-Based NMR Solution Structure: Are Solution Structure Boring?

The use of paramagnetic metals as shift and relaxation probes to determine macromolecular conformations goes back to the early steps of biomolecular NMR [111–114], about ten years before the first NMR solution structures were obtained by means of scalar and dipolar couplings [115]. During the mid 1990s, when first solution structures of paramagnetic proteins appeared, paramagnetic nuclear relaxation rates were included into currently available solution structure protocols [116] and were used as a source of NMR restraints, complementary to the information based on nuclear Overhauser effects (NOE) [117–120]. Not surprisingly, the first paramagnetic protein solved was the [4Fe–4S] cluster containing protein high potential iron–sulfur protein (HiPIP) I from *E. halophila* [121], confirming the already consolidated enrollment of iron–sulfur proteins as ideal playground for biophysicists [122]. When the paramagnetic center has anisotropic magnetic susceptibility, purely orientation-based restraints, completely independent on the distance from the metal center, opened new avenues for solution structures of proteins [123]. Residual dipolar coupling (*RDC*) arising from self-oriented paramagnetic proteins, combined with pseudo-contact shifts (*PCS*) and cross correlation rates (CCR) [124–126] succeeded to place the oriented motifs with respect to the molecular frame. Specifically tailoring algorithms provided a backbone structure in the absence of NOE measurements [127].

These achievements re-revealed to the whole biomolecular NMR community the idea that paramagnetic centers can be used as probes to determine macromolecular conformations also in diamagnetic proteins. Indeed, the use of nitroxide radicals as site-directed spin labelling in large molecular weight proteins provided paramagnetic broadening effects, which were converted into distance restraints for structure determination of large molecular mass (MM) systems [128]. The acronym PRE (paramagnetic relaxation enhancements) was coined [95] to define a novel class of structural restraints based on the metal-to-nucleus distances, derived from the paramagnetic contribution to observed relaxation rates. Since they are long range constraints, able to provide metal-to-nucleus distances up to 35 Å [129,130], PREs appeared a good alternative/complement to *RDC* in order to obtain solution structures of very high MM assemblies, where NOEs and scalar couplings are unable/insufficient to get a structure. In proteins, extrinsic paramagnetic centers can be attached via conjugation to a specific, solvent exposed, site [131], while metal chelators can be incorporated to DNA enabling PRE measurements on protein–DNA complexes [132]. The applications of PRE flourished: macromolecular structures have been characterized using PRE not only for soluble proteins [133], but also for protein–protein [134–137] and protein–nucleic acid complexes [138,139], membrane proteins [140], unfolded or partially unfolded states [141–143]. In addition, in solid-state protein NMR spectroscopy, the use of PRE as long range constraints, coupled with *PCS* [144], was capable to obtain accurate structures in the absence of conventional distance or dihedral angle restraints [145] and to provide information on the quaternary structure organization of large proteins [146]. The application of PREs goes indeed beyond their use as a structural constraint, for "static" NMR structures: in the presence of fast exchange conformational equilibria or low-populated excited states, observed PREs are population-weighted averages of the PRE rates for the major and minor species, thus unravelling structural information on transient, invisible intermediates [147]. Intermolecular PREs provide structural information on encounter complexes [148–150], inter-domain motions [151], transient protein associations [152–154], non-specific protein–DNA interactions [155], intrinsically disordered proteins [156], and drug discovery [157–159].

This brief overview shows how the growing range of applications of paramagnetism-based NMR restraints fully justify the idea that NMR structures are still far from being a consolidated protocol. Within this scenario, iron–sulfur proteins constitute paradigmatic cases to investigate the application

of novel methodologies. Attempts to replace classical NMR restraints with paramagnetism-based restraints were only partially successful [160–162]. Recent studies provided examples in which the spatial dependence of paramagnetic relaxation deviates from the $r_{MH}^{-6}$ dependence [34,35]. Intermolecular effects and the introduction of anisotropic contributions to paramagnetic relaxation [163] have been evoked to account for the sizable deviations, observed in some protein structures, between experimental distances and distances calculated from PRE values [37,164]. To further investigate these aspects, it has recently been shown that, for the small HiPIP protein PioC, a solution structure can be obtained by using only PREs [165]. The comparison between a PREs-only structure, NOEs-only structure, and the structure obtained using the combination of both type of constraints shows that the root mean square deviation (RMSD) among the three families are essentially within the RMSD of each family, thus indicating that, for proteins of small–medium size and in the absence of magnetic anisotropy, paramagnetic relaxation provides reliable constraints throughout the entire protein structure and PREs can efficiently replace NOEs in solution structure calculations [165].

## 6. Conclusions

Iron–sulfur (Fe–S) proteins play crucial roles in mammalian metabolism [166]. The study of the components of the Fe–S biogenesis machineries has led to the identification of a large number of proteins, whose importance for life is documented by an increasing number of diseases linked to these components and their biogenesis. In contrast to the chemical simplicity of Fe–S clusters, their biosynthesis in vivo appears to be a rather complex and coordinated reaction [61,167,168]. The labile nature of Fe–S cluster, their high sensitivity to oxygen, and the transient protein–protein interactions occurring during the various steps of iron–sulfur protein maturation make the structural characterization in solution particularly difficult. As we have seen in the example presented here, the presence of the paramagnetic center, for at least one of the two interacting proteins, is a potential source of restraints useful to define the relative orientation of the two proteins and, at the same time, to identify the type of cluster being present and its oxidation state [169].

Dissection of the molecular steps involved in cluster trafficking within Fe–S proteins has revealed that transfer occurs through highly conserved pathways operating in the mitochondrial matrix and in the cytosolic/nuclear compartments of eukaryotic cells; therefore, we are confident that paramagnetic NMR will further contribute to the identification of the molecular snapshots of the protein–protein interaction networks.

**Funding:** This work benefited from access to Magnetic Resonance Center (CERM) and Consorzio Interuniversitario Risonanze Magnetiche MetalloProteine (CIRMMP), the Instruct-ERIC Italy Centre. Financial support was provided by European EC Horizon2020 TIMB3 (Project 810856).

**Conflicts of Interest:** The author declares no conflict of interest.

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
