# Peer review of "Paramagnetic NMR Spectroscopy Is a Tool to Address Reactivity, Structure, and Protein–Protein Interactions of Metalloproteins: The Case of Iron–Sulfur Proteins"

_magnetochemistry, doi:10.3390/magnetochemistry6040046_

Round 1
Reviewer 1 Report
In this paper, Dr. Piccioli reviewed the recent development in using paramagnetic NMR spectroscopy to study the reactivity, structure and protein-protein interactions of iron-sulfur proteins. The review is well organized and thoroughly researched, and is an important contribution to the field of paramagnetic NMR and iron-sulfur proteins. Some minor revisions are recommended.
1) some background on the hyperfine shifts should be included in the Introduction part.
2) Ref 11 is not a good paper to demonstrate the decreased popularity of NMR. The author might want to change that.
3) line 40, 'many methodological developments have been proposed, that contributed to expand the range of applications', the author might want to give some examples of these methodological developments.
4) line 66, 'involved into' should be 'involved in'.
5) line 70-73, the sentences here need reorganizing, also references are needed.
6) ref 40 doesn't seem to fit here as it addresses ISC machinery in bacteria.
7) To my knowledge, the role of frataxin is very controversial, several functions have been proposed and no consent has been reached, and the author might want to acknowledge that.
8) The uses of holo- vs holo or apo- vs apo are not consistent.
9) the uses of [Fe2S2] vs [2Fe–2S], [Fe4S4] vs [4Fe-4S], or FeS vs Fe-S are not consistent.
10) line 102, GRX5 should be GLRX5
11) line 130-140, citations are needed here. It would be helpful to illustrate the differences between ISCA and ferredoxins with a figure.
12) 147, C-term should be C-terminal
13) line 187-195, relavent citations are needed here.
14) The uses of iron-sulfur vs Iron-Sulfur are not consistent
15) line 293, the author should point out what kind of iron-sulfur proteins were studied by Markley and coworkers using direct detection methods.
16) Abbreviations should be explained the first time it's introduced, e.g. RDC, PRE, PCS, MW, RMSD, Fe-S, etc.
17) The author might want mention the distance range of PRE constraints.
Reviewer 2 Report
This is a very detailed and well-written review of the recent progress in the field of the paramagnetic NMR with an accent on the iron-sulfur proteins. The only recommendation I have is to add additional introductory section which shortly describes the main approaches in the field of the paramagnetic NMR (PRE, RDC, PCS, contact shifts, etc.). While in the current version of the review all these concept are shortly introduced than firstly mentioned in the paper, I would argue that combining them in the introduction would help non-specialists to access this review.
The manuscript also will require a minor spell check to catch remaining typos (word "paramagentism", for example, is found is several places across the paper).
